# A Population-Based Analysis of Incidence, Mortality, and Survival in Testicular Cancer Patients in Lithuania

**DOI:** 10.3390/medicina55090552

**Published:** 2019-08-30

**Authors:** Mingaile Drevinskaite, Ausvydas Patasius, Marius Kincius, Mindaugas Jievaltas, Giedre Smailyte

**Affiliations:** 1Faculty of Medicine, Vilnius University, 03101 Vilnius, Lithuania; 2Laboratory of Cancer Epidemiology, National Cancer Institute, 08406 Vilnius, Lithuania; 3Department of Public Health, Institute of Health Sciences, Faculty of Medicine, Vilnius University, 03101 Vilnius, Lithuania; 4Department of Oncourology, National Cancer Institute, 08406 Vilnius, Lithuania; 5Urology department, Lithuanian University of Health Sciences, Medicine Academy, 44307 Kaunas, Lithuania

**Keywords:** testicular cancer, seminoma, non-seminoma, incidence, mortality, relative survival, trends

## Abstract

*Background and objectives:* The aim of this study was to analyze trends in testicular cancer incidence, mortality, and survival in Lithuania during the period 1998–2013. *Materials and Methods:* The study was based on all cases of testicular cancer reported to the Lithuanian Cancer Registry between 1998 and 2013. Age group-specific rates and standardized rates were calculated using the direct method (European standard population). The Joinpoint regression model was used to provide the annual percentage change (APC). Five-year relative survival estimates were calculated using period analysis. Relative survival was calculated as the ratio of the observed survival of cancer patients and the expected survival of the underlying general population. *Results:* During the study period, the age-standardized incidence rate of testicular cancer increased from 1.97 to 3.45 per 100,000, with APC of 2.97% (95% CI 0.9 to 5.1). Incidence rate of seminomas changed from 0.71 to 1.54 per 100,000, with APC of 2.61% (95% CI −0.4 to 5.7), and the incidence rate of non-seminomas increased from 0.84 to 1.83 per 100,000, with APC of 4.16% (95% CI 1.6 to 6.8). The mortality rate of testicular cancer in Lithuania during this period declined from 0.78 to 0.51 per 100,000, with APC of −2.91% (95% CI −5.5 to −0.3). Relative five-year survival ratio for the period 2009–2013 was 89.39% (95% CI 82.2 to 94.4). In our study, the overall five-year relative survival increased slightly (10.1%) from 2004–2008 to 2009–2013 (from 79.3% to 89.4%). *Conclusions:* A moderate increase of testicular cancer incidence has been observed in Lithuania between the years 1998 and 2013, while the mortality rate decreased. The five-year relative survival increased according to different period estimates; however, the results could have been higher if a multidisciplinary approach to diagnostics and management in the concerned centers had been implemented in Lithuania as in other countries.

## 1. Introduction

Accounting for approximately 1% of all male cancers, testicular cancer is a rare malignancy worldwide; however, testicular cancer is the most frequent malignant disease in young adult men (aged 15–45 years) [1,2]. Numerous publications have shown an increasing incidence of testicular cancer in the last 40 years, with substantial differences among countries. Testicular cancer incidence remains the highest in developed nations with primarily Caucasian populations [3,4,5,6,7]. Northern Europe is the highest testicular cancer incidence area, with the highest rates observed in Norway and Denmark [5]. The incidence of testicular cancer in Lithuania is one of the lowest in Northern Europe—2.1 cases per 100,000 people in Lithuania and 15.4 per 100,000 in Denmark were reported for the same period [8]. In the most recent analysis, the highest incidence rate was reported in Norway, with 11.5 cases per 100,000 people, whereas testicular cancer incidence in Lithuania was maintained at a low of 2.6 per 100,000 [9].

Testicular cancer constitutes a diverse group of neoplasms of which approximately 98% are testicular germ cell cancers. Testicular germ cell cancers are traditionally classified into three main types: seminomas, non-seminomas, and spermatocytic seminomas. Seminomas comprise 50–60% of all testicular cancers in the majority of countries in the different parts of the world [3]. The median age at diagnosis of seminoma is 35–39 years, whereas for non-seminoma, the median age at diagnosis is 10 years younger, at 25–29 years [10].

Testicular cancer has a largely unexplained etiology. Although seminomas and non-seminomas have different clinical characteristics, studies have revealed similar trends in incidence in the majority of countries, which may indicate that both types share common etiological risk factors [4,5]. Known risk factors for testicular cancer are cryptorchidism, a previous diagnosis of testicular cancer, a genetic predisposition, and maternal estrogen exposure [11]. Other risk factors are white ethnicity, small birth weight, preterm birth, small gestational age, inguinal hernia, twinning, subfertility, testicular dysgenesis syndrome [12,13], adult height, and low BMI (Body mass index) [14,15]—possibly as proxies of the birth-cohort effect. Somewhat less relevant factors are profession (firefighting, aircraft maintenance), environmental exposure (organochloride pesticides), and best-practice tumour management [16]. The aim of another study was to investigate marijuana use risk factors [17,18]. Many cases of control studies and some cohort studies support different etiologies of seminomas and non-seminomas; however, other studies have revealed little variation in risk factors between the two subtypes, and where particular associations have been found, they have been inconsistent across studies [19]. Despite increasing testicular cancer incidence, a marked decline in mortality has been observed in several European countries already since the mid-1970s, due to advances in chemotherapy. As of recently, it is possible to analyze histology-specific population-based incidence, mortality rates of testicular cancer, and to provide age-specific and histology-specific relative survival estimates in Lithuania.

## 2. Materials and Methods

The study is based on the Lithuanian Cancer Registry database, covering a population of around 3 million residents according to the 2011 census. The main sources of data are notifications gathered from all hospitals and diagnostic centers in Lithuania. The data from the Lithuanian Cancer Registry database is publicly available. Additionally, death certificate information and population registry information to verify vital status are available. The study was based on all cases of testicular cancer (International Classification of Disease, Tenth Revision (ICD-10) C62) reported to the Registry during 1998–2013. Patients were followed up with respect to vital status until 31 December 2017. Nine cases notified only by the death certificate (DCO) or only by the autopsy were excluded from survival analysis. For the analyses, patients were categorized by age at diagnosis (0–14, 15–49, and 50+ years), tumor, histology, and stage of the disease. The histology of tumors was coded according to ICDO-3. The histology codes were grouped into the following four categories: seminoma (ICDO-3 codes: 9060–9064), non-seminoma (ICDO-3 codes: 9065, 9070–9072, 9080–9085, 9100–9102), other, and unspecified. Stage of the disease was based on TNM classification.

Age-specific and age-standardized incidence rates were calculated. Standardization was performed using the direct method (European standard population). Age-standardized rates were calculated for all ages combined, histology categories, and age groups.

The Joinpoint regression analysis was used to model the rates and calculate the annual percentage change (APC) for incidence and mortality trends. APC is computed for each of those trends by means of generalized linear models assuming a Poisson distribution. Joinpoint analysis was performed for all ages combined and age-specific rates for the following age groups: 0–14, 15–49, and 50+ years. Joinpoint software version 4.6.0.0; Statistical Methodology and Applications Branch, Surveillance Research Program, National Cancer Institute, Bethesda, MD, USA was used [20].

Five-year relative survival estimates were calculated using period analysis, providing more up-to-date survival estimates than traditional cohort-based analysis [21]. The period analysis included only survival experience during the period from 2004 to 2013.

The relative survival was calculated as the ratio of the observed survival of cancer patients and the expected survival of the underlying general population. The latter estimate was calculated according to the Ederer II method, using national life tables for the male population stratified by single year of age and calendar year. All calculations were conducted with STATA v.11 (StataCorp LP, College Station, TX, USA); relative survival analysis was performed with the strs module.

## 3. Results

From 1998 to 2013, 622 testicular cancer cases were diagnosed among men in all age groups. In this period of 15 years, there was an average of 42 cases diagnosed per year. The majority of the patients (85.21%) were 15–49 years of age at the time of diagnosis, 13.98% of patients were 50 years of age or older, and only 0.81% were younger than 15 years. Seminoma was diagnosed among 46.78% of patients and non-seminoma among 42.28% of patients; the other histological types and unspecified cancers were diagnosed in 10.94% of patients (Table 1).

Figure 1 presents the age-standardized incidence and mortality rates of testicular cancer in Lithuania. From 1998 to 2013, incidence increased from 1.97/100,000 to 3.45/100,000, resulting in an annual percentage change of 2.97% (95% CI 0.9 to 5.1) (Table 2). Testicular cancer-caused mortality declined from 0.78/100,000 to 0.51/100,000, with APCs of −2.91% (95% CI −5.5 to −0.3).

Within the study period, the incidence rate of seminomas increased by 2.61% (95% CI −0.4 to 5.7) annually (from 0.71/100,000 to 1.54/100,000), and by 4.16% (95% CI 1.6 to 6.8) for non-seminomas (from 0.84/100,000 to 1.83/100,000) (Figure 1).

For seminomas, the highest incidence rate of 54.19 per 100,000 was observed in the 35–39 years age group, whereas the incidence was only 15.74 in the 45–49 years age group for the period 1998–2013 (Figure 2). For non-seminomas, the highest incidence was in the 25–29 years age group, with 51.48 per 100,000.

Table 2 presents the age-standardized and age-specific incidence trends of testicular cancer in Lithuania during the period of 1998–2013. The highest age-specific incidence increases were among adolescents and men aged 15–49 years. The incidences of seminomas and non-seminomas increased, with APCs of 3.83% (95% CI 0.6 to 7.2) for seminomas and an APC 4.75 (95% CI 2.3 to 7.3) for non-seminomas. In men older than 50 years, the total incidence rate of testicular cancer was decreased, with APCs of −4.23% (95% CI −8.2 to −0.1).

Table 3 presents the five-year relative survival for testicular cancer in Lithuania from 2004–2013. The total five-year relative survival for testicular cancer was 89.39% during the period from 2009–2013. Survival varied by age, with increasing age being associated with reduced relative survival. Slightly higher survival was observed in the seminoma group, with 93.98% survival (95% CI 80.9 to 100), than in the non-seminoma group, with 87.94% survival (95% CI 77.3 to 94.2). By TNM summary stage, the five-year relative survival was 94.36% for localized tumor stage I, 97.23% for stage II, 64.77% for stage III, and 54.67% for unknown stage.

The overall five-year relative survival increased by 10% from 2004–2008 to 2009–2013. Positive changes in survival were evident in the 15–49 years age group and for the main histological groups and all disease stages.

## 4. Discussion

In our study, we analyzed incidence trends of testicular cancer, mainly focusing on histology (seminoma and non-seminoma), and histology-specific relative survival estimates in Lithuania during the period from 1998 to 2013. We observed similar incidence and mortality changes to those reported in other Western countries. In addition, during the study period, increasing survival rates were observed.

The incidence rate of testicular cancer is continuing to increase in many populations worldwide [3,7]. The increase is most notable among European populations, with the steepest increase of incidence in the Northern European countries of Norway (age-standardized rate: 11.5 cases/100,000 people) and Denmark (10.2/100,000) [9]. According to the most recent article by Gurney et al., the rates of change in the highest-incidence countries of Norway (2.4% (95% CI 2.0 to 2.8) and Denmark (0.8% (95% CI 0.4 to 1.3) were more modest over the follow-up period, with APCs nearing 0% in more recent decades. The majority of countries from regions with relatively low testicular cancer incidence experienced a 3–5% average annual growth in incidence rates, including all Central/Eastern European countries [9]. Similarly to other countries, we observed an incidence increase in Lithuania (3.45/100,000); however, the results were relatively lower compared to Northern European countries, and the APC was slightly higher (2.97% (95% CI 0.9 to 5.1)).

The incidence of seminomas tends to be greater than the incidence of non-seminomas, with both histological types increasing over time [9]. The proportion of seminomas compared to non-seminomas in Lithuania is 46%, which is slightly lower than in other European countries (55–60%) [22,23]. Rates of seminomas and non-seminomas tended to have similar trends over time; however, there was some evidence of divergence over the last decade, wherein the incidence of seminomas appears to be increasing at a greater rate than that of non-seminomas in some higher-incidence countries [9]. The highest-incidence proportions of seminoma during the period from 1998–2003 were observed in Germany (65%), Italy (62%), and Switzerland (61%) [22,24]. Detailed analyses of the incidence of seminomas and non-seminomas in several European countries showed increasing rates of non-seminomas or seminomas in the majority of countries [8,22]. In our study, we observed a constant incidence increase of seminomas, with APCs of 2.61%, and non-seminomas, with APCs of 4.16%, which indicates there is no difference when compared to most European countries.

The observed lag of approximately 10 years in age at the peak incidence of subtypes has been constantly observed in Western populations, with non-seminomas peaking earlier (at about 25–29 years) than seminomas (at about 35–39 years) [9,25]. In our study, the highest incidence was in the 35–39 years age group, with 54.19 cases per 100,000 people, whereas for non-seminomas, the highest incidence was in the 20–29 years age group, with 51.48 per 100,000. The different age profiles may reflect that non-seminomas are more aggressive and rapidly growing than seminomas at diagnosis, and the proportion of metastatic to localized tumors is often higher for non-seminomas than seminomas [10].

It is debatable whether the observed incidence increase of testicular cancer could be partially explained by an increase of cryptorchidism incidence, one of the best-established risk factors for testicular cancer [24]. However, the detection of an increase of cryptorchidism is hampered by the methodological flaws of many monitoring systems [24,26]. Some studies state that an increase in the prevalence of cryptorchidism could not explain the considerable increase of the incidence of testicular cancer [8,22].

The estimated age-standardized five-year relative survival according to EUROCARE-5 is 88.6% in Europe [23]. Age-standardized five-year relative survival was highest for testicular patients from Northern Europe (92.8%), followed by 91.8% for patients from Central Europe, 89.1% for patients from Southern Europe, and 80.1% for patients from Eastern Europe. Age-standardized five-year relative survival in Lithuania is 67%, according to EUROCARE-5. In our study, overall five-year relative survival increased slightly (10.1%) from 2004–2008 to 2009–2013 (from 79.3% to 89.4%).

The survival for patients with seminoma was always higher than that for non-seminomas in Europe and in all the European regions [23]. Overall five-year relative survival in Europe for seminomas and for non-seminomas was estimated to be 93.9% and 88.3%, respectively. The highest five-year relative survival for seminomas and non-seminomas was observed in Northern European countries (97.7% and 90.2%) [23]. In a study by Stang et al., five-year relative survival for seminomas was found to be close to 100% (97.6%), and the difference was considerably noticeable compared to non-seminomas (93.3%) [27]. In our study, a five-year relative survival was still slightly lower compared to other countries: 93.98% for seminomas and 87.94% for non-seminomas.

Empirical evidence of the age-dependency of the prognosis of testicular cancer goes back at least to 1943 [28]. In our study, the estimated five-year relative survival was better in the younger age groups and decreased with increasing age. Spermon et al. compared relative survival in men younger and older than 50 years with testicular cancer. The 10-year relative survival was calculated for both patient groups, and patients younger than 50 years had a better 10-year relative survival (90.8%) than those who developed the cancer after the age of 50 years (84%) [29]. According to EUROCARE-5 results, the five-year relative survival was 95% in patients of 15–39 years and 90% in those aged more than 40 years [23].

A recent detailed study was able to represent the independent effect of age on the prognosis of testicular cancer [30]. In a study by Fossa et al., an important finding was noted: the adverse impact of increasing age on testicular cancer-specific mortality, while taking into account disease characteristics, treatment factors, and sociodemographic variables. They hypothesized that reduced treatment intensity combined with increased therapy-related toxicity is a plausible explanation for increased testicular carcinoma-specific mortality in patients older than 40 years [30]. Hoffman et al. showed that low sociodemographic status reduced the likelihood of being offered radiotherapy [31,32]. In their study, men residing in countries with a higher education level were more likely to receive adjuvant radiotherapy for stage I seminoma than men residing in countries with a lower education level, which leads to reduced active surveillance [31]. Active surveillance avoids unnecessary treatment and the risk of treatment-related adverse effects [33].

Differences in five-year relative survival between our study and other Northern European countries is a cause of concern. Possible factors explaining these differences could be: diagnostic uncertainty due to the lack of tools for the diagnosis, limited access to appropriate therapies, delayed treatment, or lack of evidence-based guidelines [34]. Testicular cancer is a rare malignancy that comprises only 1% of all cancer worldwide. In Lithuania, due to the rarity of this disease, we get approximately 40 patients per year, who are treated in different centers. Centers have their own diagnostic and treatment guidelines, with most of them following European Association of Urology guidelines for testicular cancer. Sandrucci et al. and Blay et al. explained the wide consensus among rare cancer experts, that patients with rare cancers should be treated at centers from the beginning of their clinical history with a multidisciplinary clinical decision on how to plan treatment and the quality of the initial surgical intervention, which can largely determine patients’ outcomes [34,35]. A good example to be followed for the multidisciplinary approach is that of the Swedish and Norwegian Testicular Cancer Group (SWENOTECA), a collaboration that resulted in excellent patient outcomes [36].

The mortality of testicular cancer has declined over time, despite the incidence increase of testicular cancer [24]. The introduction of cisplatin-based therapies in the late 1970s brought about a decrease in mortality rates, with survival rates reaching 95% [37]. It has been stated that the decline of testicular cancer mortality in Eastern European countries started in the 1980s, but at a rate slower than that recorded in Western Europe [38]. Mortality rate in Lithuania decreased as well, with APCs of −2.91%, suggesting beneficial effects of a prompt diagnosis followed by effective early diagnosis, treatment, and surveillance. It should be noted that increasing awareness of self-examination and generational shift in attitudes to men’s health contributes to early diagnosis and better outcomes [39].

The reliability of our results depends on the quality of cancer registration. Changes in completeness of registration over time may have artificially influenced temporal trends to some extent. Furthermore, a certain degree of misclassification between seminomas and non-seminomas as well as between germ cell tumors and non-germ cell tumors has most likely occurred. If the level of misclassification has changed over time, separate analysis of the histologic subgroups may have produced altered trend appearances [8]. Despite the limitations of our study, this nationwide analysis highlights changes in testicular cancer incidence, mortality, and survival, indicating an existing gap for survival improvement.

## 5. Conclusions

The incidence and mortality trends of testicular cancer is continuing to change in Lithuania as well as in other European countries. Five-year relative survival increased according to different period estimates; however, it does not reach the highest scores of to Northern European countries. Taking into account the experience of other countries, survival improvement of testicular cancer patients could be reached with a multidisciplinary approach to diagnostics and management in dedicated centers, involving precise staging, adequate treatment, and attentive follow-up.

## Figures and Tables

**Figure 1 medicina-55-00552-f001:**
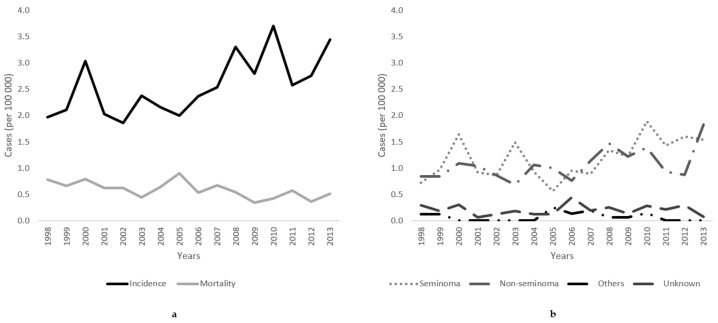
(**a**) Age-standardized incidence and mortality trends and (**b**) incidence rates of histology-specific testicular cancer (seminoma, non-seminoma, other, and unspecified) of testicular cancer in Lithuania from 1998 to 2013.

**Figure 2 medicina-55-00552-f002:**
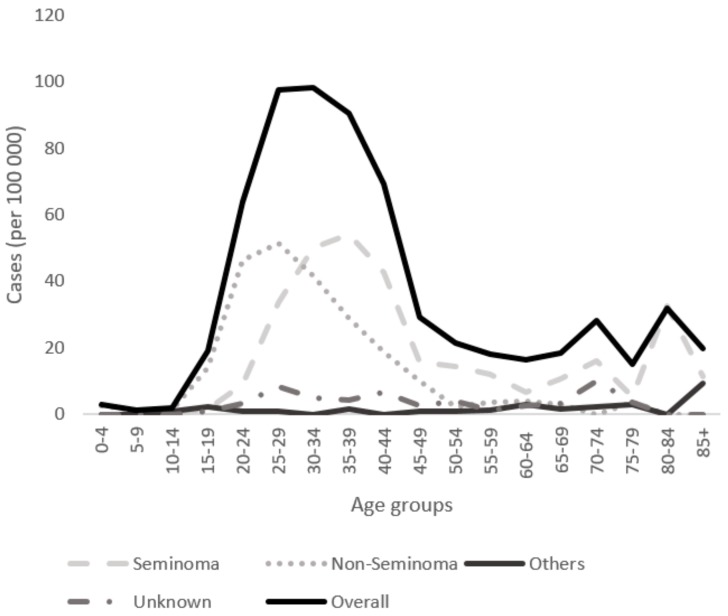
Age-specific incidence rates of histological types (seminoma, non-seminoma, other, and unspecified) for the period from 1998 to 2013 in Lithuania.

**Table 1 medicina-55-00552-t001:** Study group characteristics by age and histology in Lithuania during the period of 1998–2013.

	Seminoma		Non-Seminoma		Others		Unknown		Total	
Age Groups (Years)	Cases	%	Cases	%	Cases	%	Cases	%	Cases	%
0–14	0	0	4	1.52%	1	5.90%	0	0.00%	5	0.81%
15–49	238	81.79%	248	94.29%	8	47.05%	36	70.59%	530	85.21%
50+	53	18.21%	11	4.19%	8	47.05%	15	29.41%	87	13.98%
Total	291	100.00%	263	100.00%	17	100.00%	51	100.00%	622	100%
Stages										
I	166	57.04%	129	49.05%	6	35.29%	12	23.53%	313	50.32%
II	76	26.12%	52	19.77%	3	17.65%	5	9.80%	136	21.86%
III	37	12.71%	64	24.33%	3	17.65%	11	21.57%	115	18.50%
Unknown	12	4.13%	18	6.85%	5	29.41%	23	45.10%	58	9.32%
Total	291	100%	263	100.00%	17	100%	51	100%	622	100%

**Table 2 medicina-55-00552-t002:** Age-standardized incidence trends of testicular cancer by histology type in Lithuania during the period of 1998–2013.

	All Ages				15–49 Years				50+ Years			
	ASR*		APC**	95% CI	ASR		APC	95% CI	ASR		APC	95% CI
	Years				Years				Years			
	1998	2013			1998	2013			1998	2013		
Any testicular cancer	1.97	3.45	2.97*	0.9 to 5.1	1.45	2.78	3.98*	2.1 to 5.9	0.49	0.45	−4.23*	−8.2 to −0.1
Seminoma	0.71	1.54	2.61	−0.4 to 5.7	0.56	1.07	3.83*	0.6 to7.2	0.14	0.38	−2.42	−6.8 to2.2
Non-seminoma	0.84	1.83	4.16*	1.6 to 6.8	0.72	1.64	4.75*	2.3 to 7.3	0.07	0.06		
Other	0.12	0			0.05	0			0.03	0		
Unspecified	0.29	0.07	0.14	−4.6 to 5.1	0.11	0.07	1.89	−5.6 to 10.0	0.21	0		
Mortality	0.78	0.51	−2.91*	−5.5 to −0.3	0.33	0.28	−3.05	−6.0 to −0.0	0.49	0.19	−4.27	−8.3 to −0.1

*ASR - age-standardised rate.**APC - annual percentage change.

**Table 3 medicina-55-00552-t003:** Five-year relative survival for testicular cancer in Lithuania from 2004–2013.

	Period Estimates	
	2004–2008	2009–2013
Total	79.32 (71.72–85.39)	89.39 (82.21–94.38)
Age at diagnosis (years)		
0–14	100 (100–100)	100 (100–100)
15–49	82.19 (74.39–88.04)	91.32 (84.18–95.77)
50+	55.19 (29.10–79.12)	20.85 (37.59–96.23
Histology		
Seminoma	77.04 (64.38–86.50)	93.98 (80.91–100)
Non-seminoma	84.67 (73.37–91.84)	87.94 (77.34–94.23
Other	72.33 (31.20–94.76)	90.37 (17.64–100)
Unspecified	66.25 (34.73–86.58)	57.72 (23.52–82.28)
Stage at diagnosis		
I	95.79 (85.73–100)	94.36 (83.46–99.94)
II	76.40 (60.69–87.07)	97.23 (81.76–100)
III	52.15 (33.78–68.27)	64.77 (45.22–79.38)
Unknown	74.41 (46.23–90.70)	54.67 (12.04–85.03)

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
