# Peer review of "A Population-Based Analysis of Incidence, Mortality, and Survival in Testicular Cancer Patients in Lithuania"

_medicina, 2019, doi:10.3390/medicina55090552_

Round 1

Reviewer 1 Report

Table 1,2,3 should be re-organized according to narrower age intervals. For example, in line 122-124, the author mentioned incidence rate in the age 35-39 year age group, 45-49 year age group, 25-29 year age group, respectively. However, such detailed information cannot be found in all Tables which only show 15-49 age group including teenage, youth, middle-aged group. This is not a very clear and scientific age group classification method which should changed into narrower age intervals corresponding to the text part of manuscript because Tables are direct and most important information source for readers and should be accurate and convenient for understanding the research aim of the present topic. 

Reviewer 2 Report

Thank you for inviting me to review this very well written manuscript. Below are a few comments in order to improve this paper.  

1) This is a suggestion, I recommend abbreviating testicular cancer (TC or TCa) since it is a recurring term in this paper

2) Page 2, Line 50: the sentence "Testicular cancer constitutes a diverse group of neoplasms..." should be on a new paragraph

3) To ensure that all risk factors are covered, I advise you to refer and cite these two papers:

See table 2 in: https://doi.org/10.22374/ijmsch.v2i1.16

See table 2 in DOI: 10.1177/1557988315626508 

4) Page 2, Line 75 and 76: Additionally, death certificate information, and population registry information to verify vital status ARE available (ARE not IS available)

5) Page 3, Line 103: Avoid starting a sentence with a number. I recommend: The majority of patients (85.21%) were....

6) Discussion and conclusion: Have a look at this study in the UK, explaining the detection of TC at earlier stages (despite increased incidence): https://doi.org/10.1093/pubmed/fdw014 

Also, although not directly relevant to this paper, increasing men's awareness of diseases of the testes may lead to detection at earlier stages. Read the work of Saab et al. FYI:

DOI: 10.1097/NNR.0000000000000303

https://doi.org/10.1007/s10055-018-0368-x
